# Monitoring Tribological Events by Acoustic Emission Measurements for Bearing Contacts

**Philipp Renhart** [1,*], **Michael Maier** [1], **Christopher Strablegg** [1], **Florian Summer** [1], **Florian Grün** [1] **and Andreas Eder** [2]

1 Department Product Engineering, Chair of Mechanical Engineering, Montanuniversität Leoben, Franz-Josef-Straße 18, 8700 Leoben, Austria; michael.maier@unileoben.ac.at (M.M.); christopher.strablegg@unileoben.ac.at (C.S.); florian.summer@unileoben.ac.at (F.S.); florian.gruen@unileoben.ac.at (F.G.)
2 High Tech Coatings GmbH, Dr.-Mitterbauer-Str. 3, 4655 Vorchdorf, Austria; andreas.eder2@miba.com
* Correspondence: philipp.renhart@unileoben.ac.at; Tel.: +43-3842-402-1464

**Abstract:** The measurement of acoustic emission data in experiments reveals informative details about the tribological contact. The required recording rate for conclusive datasets ranges up to several megahertz. Typically, this results in very large datasets for long-term measurements. This in return has the consequence, that acoustic emissions are mostly acquired at predefined cyclic time intervals, which leads to many blind spots. The following work shows methods for effective postprocessing and a feature based data acquisition method. Additionally, a two stage wear mechanism for bearings was found by the described method and could be substantiated by a numerical simulation.

**Keywords:** acoustic emission; friction; tribology





## 1. Introduction

Since decades the measurement of acoustic emission (AE) is widespread used for machine diagnostics and condition monitoring of various mechanical systems [1]. These elastic waves in solid bodies can be detected by suitable sensors on the surface of components [2]. The evaluation methods were developed systematically since decades for various applications. In a tribological context, Hase et al. [3,4] found out that specific defects and wear mechanisms of machine components can be detected by changes of their characteristic signal properties, such as frequency and amplitude during operation. Mirhadizadeh et al. found a correlation between the AE RMS and fluid friction [5]. Nagata et al. [6] noticed the correlation between the root-mean-square (RMS) value and the surface condition, additionally a high sensitivity was found. The RMS value of a time domain signal is frequently used for diagnostics of roller bearings [7,8] and journal bearings [9]. Loutas et al. published various parameters in time- and frequency domain, in tests with artificially induced gear cracks derived parameters and various combinations were discussed in detail [10]. In the work of Eftekharnejad et al. a naturally degraded roller bearing was analyzed by use of a kurtogram [7], in [11] artificial surface defects were studied for gears. Roller bearings with seeded defects were studied by Van Hecke et al. [12]. Dwyer [13] established the spectral kurtosis (SK) firstly as a supplementary tool to the power-spectral-density (PSD) [14]. The SK is nowadays an approved concept for machine diagnostics and live monitoring of acoustic and vibration in many fields [15–18].

The observation frequencies of acoustic emission measurements is usually very high, which leads inevitable to immense datasets. This circumstance inhibits long-term recordings in relevant experiments. The measurement in intervals lowers memory requirements, with the drawback of information losses. In this work, the combination of the kurtosis and the RMS value of signals is proposed to be suitable for an event based measurement trigger.

## 2. Experimental Details

The experiments for this study were performed on a TE92HS, which was manufactured by Phoenix Tribology Ltd. (Basingstoke, UK). Major components of the TE92HS are shown in Figure 1. The test rig consists of a rigid frame with a top mounted synchronous motor, whereby the driving shaft is pivoted with high precision ball bearings for superior concentricity properties. In order to enable electrical measurements between the shaft and the specimen an isolating coupling is mounted between the drive and the shaft. Two vertical columns of the frame guide a movable beam, which carries the bearing-segment-tester (BST). Between the beam and the BST is a rotatable mounting to facilitate the measurement of the torque about the vertical axis. The force of the bellows cylinder is amplified by two transmission levers by a factor of five.

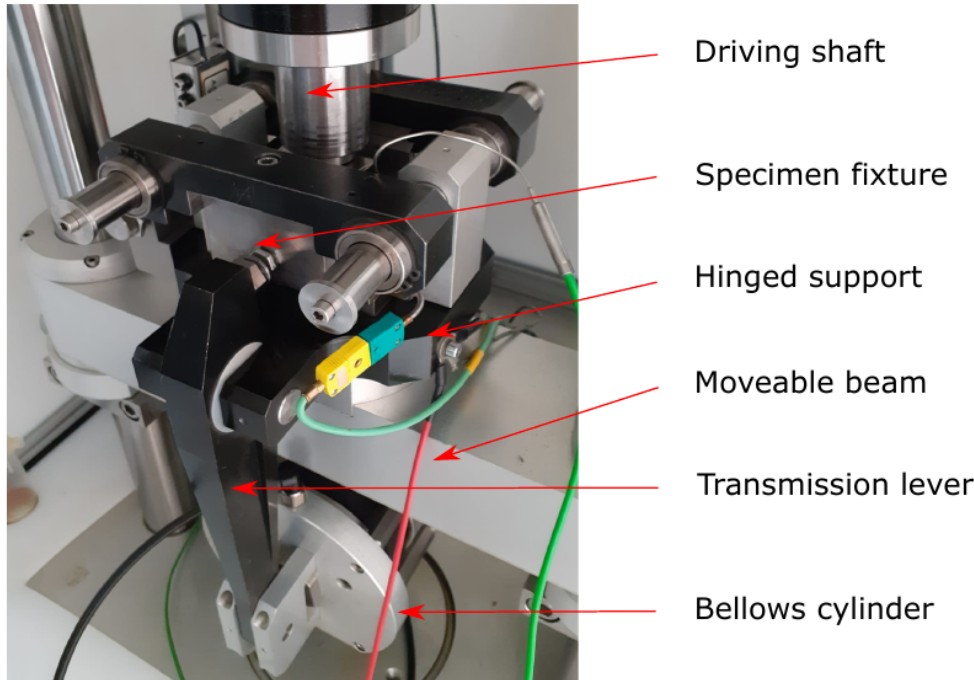

**Figure 1.** Major components of the TE92HS test rig with the bearing-segment-tester mounted.

The force of the transmission levers is induced symmetrically to both specimen fixtures and represents the controlled normal force during a test. The main components of the bearing-segment-tester are shown in Figure 2. In the center of the BST, the shaft specimen is rotating at a certain speed, simultaneously both bearing segments are pressed against the shaft laterally. The width of the surface in contact is 6 mm and depended on the design of the 120° bearing segments. On the rear specimen fixture a bracket was mounted for the AE sensor (Type 8152C0000320, Kistler Group, Winterthur, Switzerland), this protects the sensor of overheating and moving parts. The position of the sensor was chosen as close as possible to the contact and with a distinct space to the oil. In order to reduce signal damping the bracket is tightly screwed to the fixture with only one joining surface. The BST is located in a heatable basin, which contains up to 80 mL of lubricant. Once the full amount of oil is filled in the contacting surfaces are fully immersed.

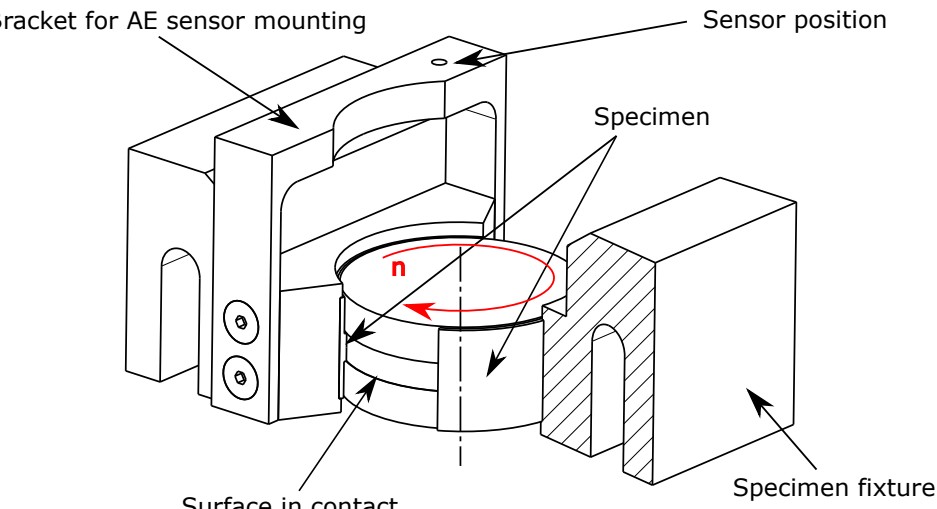

**Figure 2.** The main components of the bearing-segment-tester. Two bearing segments are clamped in the specimen fixtures and pressed against the rotating shaft. The width of contacting surface is 6 mm and is highlighted on the shaft specimen. For the AE sensor an additional sensor mount is attached to the rear specimen fixture.

*2.1. Test Programme and Measurements*

The Start-Stop-test (SST) is intentionally designed to determine the cyclic degeneration of a journal bearing due to repeated starts and stops. During the start up of a journal bearing the coefficient of friction changes with increasing relative speed between the shaft and the bearing. The absolute value of the COF is the sum of friction between the solid parts and hydrodynamic friction. With increasing relative speed hydrodynamic friction gains in significance and the lubrication gap height becomes larger with the effect that friction between solid parts decreases. As the relative speed increases further shear forces in the lubricant increase in importance resulting in a rise of the COF. These interdependencies are usually shown in the stribeck curve [19]. The SST starts with a defined running in routine of 30 min at a constant speed of 100 rpm and constant load at the same time the oil bath is heated up with a linear temperature gradient. All tests in this work were performed at 1 MPa nominal load and at 100 °C oil temperature. As soon as the temperature is constant the spindle stops and the speed cycling begins. The shaft accelerates constantly, which results in a linearly accelerated rotary motion up to a top speed of 500 rpm within 15 s. After the shaft reaches the top speed, it decelerates within a few seconds to 0 rpm. This cycle is repeated 9000 or 4500 times. Identical SSTs were performed on the same test rig in [20,21].

On the TE92HS the sampling rate is set to 10 Hz for the continuous measurement channels such as frictional torque (0–6 Nm), temperatures (up to 250 °C), contact potential (0–50 mV), acoustic emission RMS (0–10 V), normal force (0–2.5 kN) and speed (0–500 rpm). For the detailed evaluation of the stribeck curve every hundredth cycle is measured with 1 kHz with the so called high-speed measurement system. However, this recording starts at the very beginning of relevant cycles. In parallel, a binary trigger signal is set and the data acquisition of the raw AE signal starts at the NI USB-6361 A/D converter with a sampling rate of 850 kHz. A detailed description of the measurement chain and the triggering is given in [20].

The investigated journal bearing segments for this work are magnetron sputtered Ag coating on a 34CrNiMo4 steel back with an adhesion promoting layer (APL) inbetween, as published previously [20]. The initial arithmetic surface roughness of the shaft is tolerated to 0.16 μm. The initial surface of the unused bearing segment showed an arithmetic surface roughness of 0.21 μm.

### 2.2. Data Processing of Raw AE Signals

For this work, a $1/n$-octave filter bank with $n = 12$ was used. The standardized frequency range [22] was extended systematically. In order to use the entire measuring range of the AE-sensor, the highest frequency $f_2$ was chosen to be 425 kHz. The relation between the higher frequency $f_2$ and the lower frequency $f_1$ of a band was calculated with Equation (1).

$$f_2 = \sqrt[n]{2} \cdot f_1 \tag{1}$$

The center frequency $f_c$ of a single octave band is the geometric mean calculated according to Equation (2).

$$f_c = \sqrt{f_1 \cdot f_2} \tag{2}$$

This approach for choosing frequency ranges is inspired by EN ISO 266:1997. The relative bandwidth $b$ is in contrary to the absolute bandwidth independent from the frequency and can be calculated by Equation (3).

$$b = \frac{f_2 - f_1}{f_c} \Rightarrow b = \frac{\sqrt[n]{2} - 1}{\sqrt[n]{2}} \tag{3}$$

The absolute bandwidth in relation to the center frequency $f_c$ for different octave bands is shown in Figure 3.

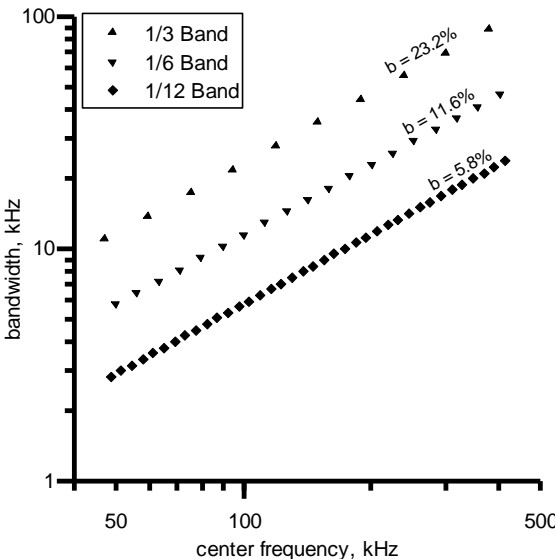

**Figure 3.** The absolute bandwidth of different $1/n$-octave bands over the investigated frequency range. Relative bandwidths are only dependent of the octave band.

All discrete AE-signals $y(i)$ were initially filtered and separated by the previously described 1/12-octave filter bank. For a fast overview the datasets were processed directly by calculating the RMS value $y_{RMS}$, the kurtosis $\gamma_4$ and the product of both $P_5$. Since the performed experiment is non-stationary in a further step the filtered signal was subdivided into 10 segments, followed by the evaluation of the statistical parameters. This procedure enables a better visibility of the transient acoustic emission signal. A schematic overview of the postprocessing method is given in Figure 4. Depending on the octave resolution $n$, the frequency range, the number of discrete time steps $t$ and data vectors $k$, the procedure leads to a matrix with the size of $i \times m \times k$.

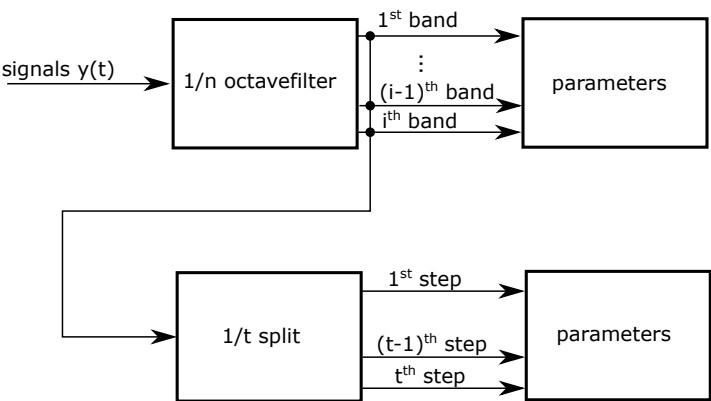

**Figure 4.** Postprocessing of the raw acoustic emission signal with an $1/n$-octave filter bank and an optional discrete subdivision in time domain.

*2.3. Simulation*

The resulting force on a rotating journal bearing is the vector sum of hydrodynamic and solid contact forces. Our numerical simulation is based on the work of Bergman et al. [23], which was a simplified model of the bearing-segment tester. The simulation was set up in the commercial FEM software Comsol Multiphysics® 5.6 to study the impact of local wear on the Stribeck curve. The calculation of the hydrodynamic forces is performed on the bearing segment with use of the Reynolds equation, which describes hydrodynamic effects in thin films accurately. The solid contact forces are calculated with a contact model, which is based on the work of Greenwood and Tripp [24]. In Figure 5a, the artificial groove with a varying depth of $w_z$ is sketched in this area, due to the stationary running-in process in mixed friction the shape of the groove width is supposed to have the same radius as the shaft. The dimension of the shaft was measured to be ∅47.432 mm averaged over 5 measuring points with a standard deviation of 0.63 µm. Boundary conditions, the external load $F_N$ and the reaction force $F_x$ of the elastic foundation is shown in Figure 5b. The elastic boundary condition in x-direction facilitates a small lateral displacement such as implemented in the test rig.

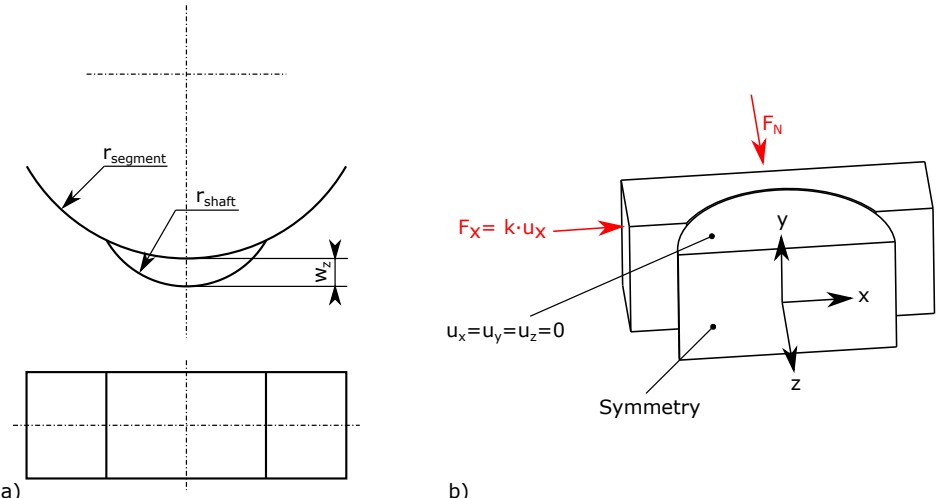

**Figure 5.** (**a**) A not to scale schematic representation of the artificial groove with the maximal depth $w_z$ for the simulation. (**b**) The numerical model with the external load $F_N$, symmetry condition and the load dependent linear boundary condition along the *x*-axis.

Further model parameters for the simulation of the bearing segments are given in Table 1. For the static friction value $\mu_0$ the averaged initial friction value of 9000 stribeck cycles was taken.

**Table 1.** Parameters defining the numerical simulation of the bearing segment.

| Description | Parameter | Value | Description | Parameter | Value |
|---|---|---|---|---|---|
| Load | $F_N$ | 249 N | Static friction | $\mu_0$ | 0.125 |
| Clearance | Ψ | 1.6‰ | Kin. viscosity | $\nu\ (T = 100°)$ | 5 mm²/s |
| Speed range | n | 0–500 rpm | Density | $\rho\ (T = 100°)$ | 0.940 kg/m³ |
| Stiffness | k | 20 kN/m | Groove | $w_z$ | 0–15 µm |

## 3. Results

### 3.1. Test Results

The parameter $P_5$ was evaluated as before, for a 9000 cycles Start-Stop-Test. In Figure 6a–j, the evolution of $P_5$ in a 1/12-octave band is shown at rising average speeds during all Stribeck cycles. At very low speeds usually boundary friction is observed, see Figure 6a. Despite the low total friction energy the 86.8 kHz band is visible permanently in all cycles. The first conspicuous value was detected in cycle 5200 at 65 kHz with an absolute value of 0.05. The analysis of this event **A** revealed a short pulse as soon as the rotation starts. Two tribological events, with a high value of $P_5$ over the full spectrum were detected at an average speed of $n_{AVG} = 4.3$ rpm. Event **B** occurred in cycle 6900 with a peak value of 14.02 in the 68.9 kHz octave band, higher frequencies showed smaller values. In cycle 8100 event **C** was observed with an identical characteristic frequency distribution, but a lower peak of 6.08 at 68.9 kHz. In Figure 6b, a single event **D**, was found in cycle 1600 with a peak of 1.22 at 51.6 kHz. Beside this, events persistent frequency bands are dominant at 86.8 kHz, 100 kHz and 138.1 kHz with peak values of $P_5 > 0.1$ visible. Tending towards higher average speeds, certain frequencies become clearer. The 86.8 kHz band is still dominant up to an average speed of 122 rpm (see Figure 6a–d). Beginning in Figure 6e, the dominance of this frequency band decreases, contemporaneously, a triplet of frequency bands stands out at 231.7 kHz, 206.5 kHz and 183.9 kHz. Independently of the frequency band, higher values of $P_5$ are found above cycle 5000, especially at higher speeds as shown in Figure 6f–j. Concluding the observations, we can state that certain frequency bands are dominant for boundary friction and decrease with upcoming hydrodynamic operation of the journal bearing. All detected major events (**A**,**B**, **C**, **D**) occurred at low average speeds.

Event **A** was found at very low speeds with $P_5 = 14.02$ at 68.9 kHz, despite this peak, the event was detected in all frequencies. A detailed view at the unfiltered raw signal reveals a high peak after about 0.3 s, which is concurrent with the beginning rotation of the spindle. In Figure 7a, a section of the raw AE signal is shown. Before the peak occurs only noise with a small steady component can be seen. As the spindle actuates, a peak amplitude of 2.18 V can be seen, followed by a decay to a steady amplitude of about 0.17 V within 3 ms. Event **B**, in Figure 7b, is quite similar to the previously described event **A**. After 0.3 s a peak value of 2.05 V was fond with a subsequent decay of the signal. A satisfying explanation could be a local adhesive contact between the steel shaft and the bearing specimen. The fact, that both events occurred in late cycles and silver has a very low adhesion tendency to steel [25] indicates that the steel shaft is partially in contact with the APL. Generally, the adhesion force is higher for smooth surfaces. An other indicator is the previously described observation, that the general behavior changes after about 5000 cycles in the entire spectrum. Hase et al. [3] observed similar burst like waveforms for a pure Cu/Fe tribosystem without lubrication during adhesive wear, despite the similarity their event lasts only about 5 µs.

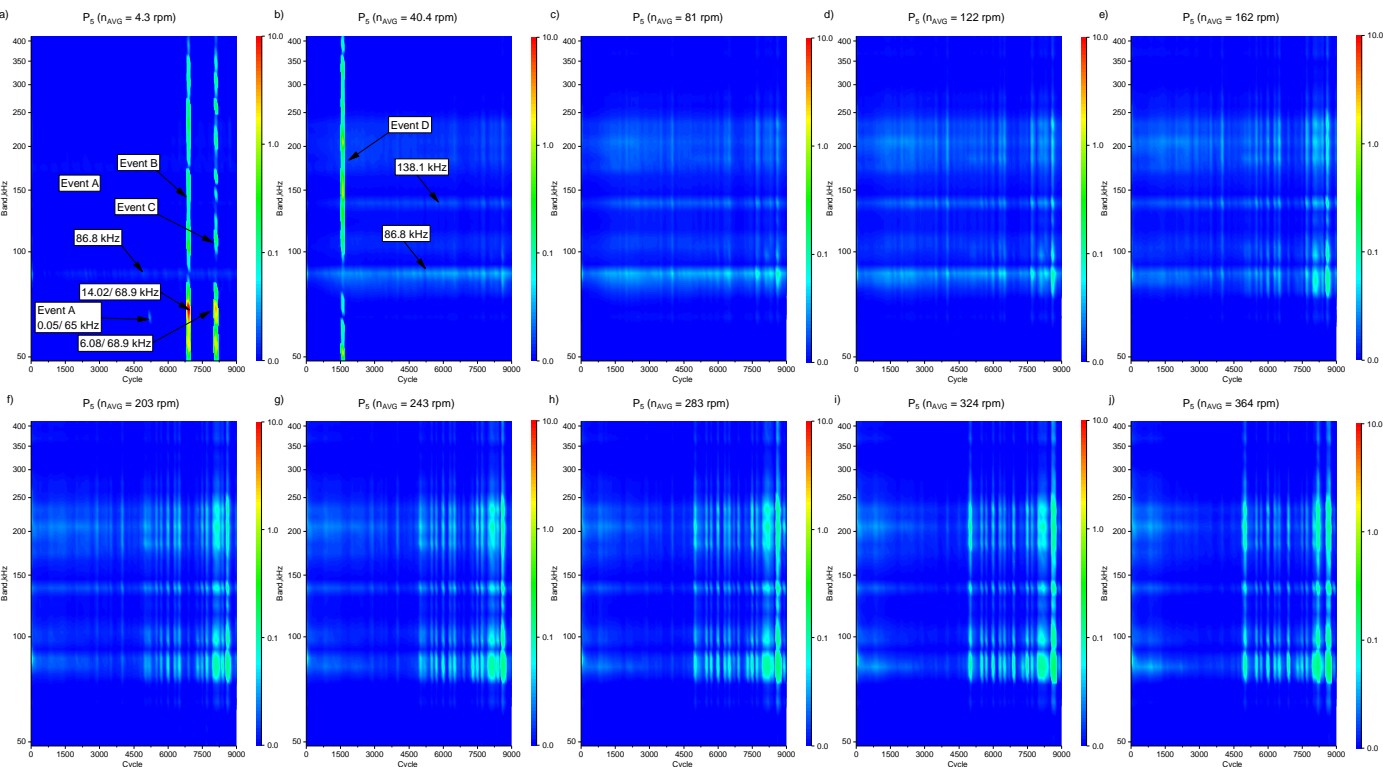

**Figure 6.** The evolution of $P_5$ over 9000 Start-Stop cycles in different frequency bands and ascending average speeds. (**a**) 4.3 rpm, (**b**) 40.4 rpm, (**c**) 81 rpm, (**d**) 122 rpm, (**e**) 162 rpm, (**f**) 203 rpm, (**g**) 243 rpm, (**h**) 283 rpm, (**i**) 324 rpm, (**j**) 364 rpm. Three tribological events were detected at speed below $n_{AVG} = 40.4$ rpm. The 86.8 kHz and the 138.1 kHz frequency band are dominant in boundary friction. Higher frequency bands as 231.7 kHz, 206.5 kHz, 183.9 kHz and 100 kHz increase proportionally with hydrodynamic effects at higher speeds.

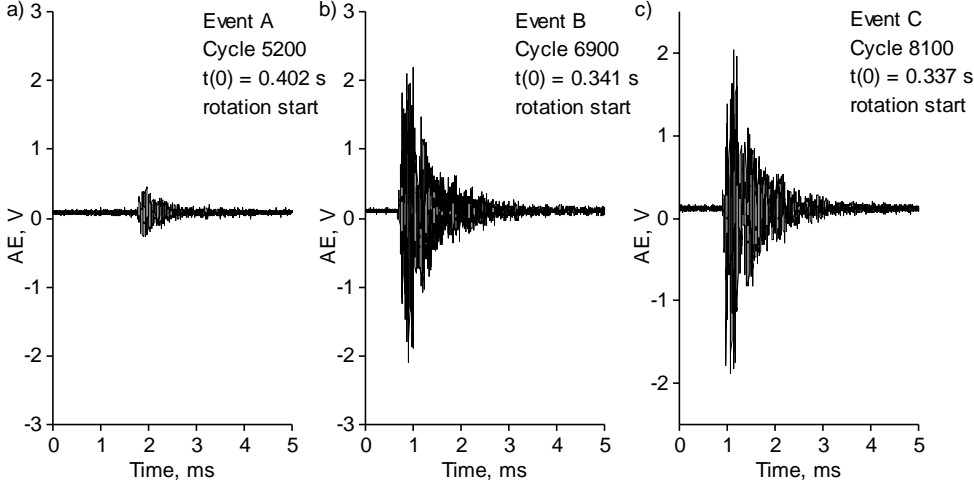

**Figure 7.** Unfiltered sections of the AE signal in the moment of the beginning rotation. In the first event, the signal decays within 1 ms, in the last of both events, the signal decays within 3 ms to a stable amplitude. (**a**) Cycle 5200, (**b**) Cycle 6900, (**c**) Cycle 8100.

Event **C** was detected in cycle 1600 with $P_5 = 1.22$ in the 51.6 kHz frequency band. The 819 µs lasting event occurred after 2.06 s at a spindle speed of about 51 rpm. A closer look at the raw acoustic emission data reveals a number of single peaks within a very short time. In Figure 8a, all peaks are clearly visible. A subsequent time dependent frequency analysis, as shown in Figure 8b, was performed for a deeper insight. The continuous wavelet transformation (morse wavelet) was used for this investigation, due to the short-

ness of the event and the sampling frequency the resolution in the frequency domain is limited. However, the frequency spectrum of the first peaks (up to 600 μs) is quite similar with a maximal amplitude at 172.1 kHz and 184.5 kHz. These resemblances are also reflected in the shape of the peaks in time domain. After about 800 μs, the peaks in time domain become wider. In the frequency spectrum, two frequencies stand out at 211.9 kHz and 139.8 kHz.

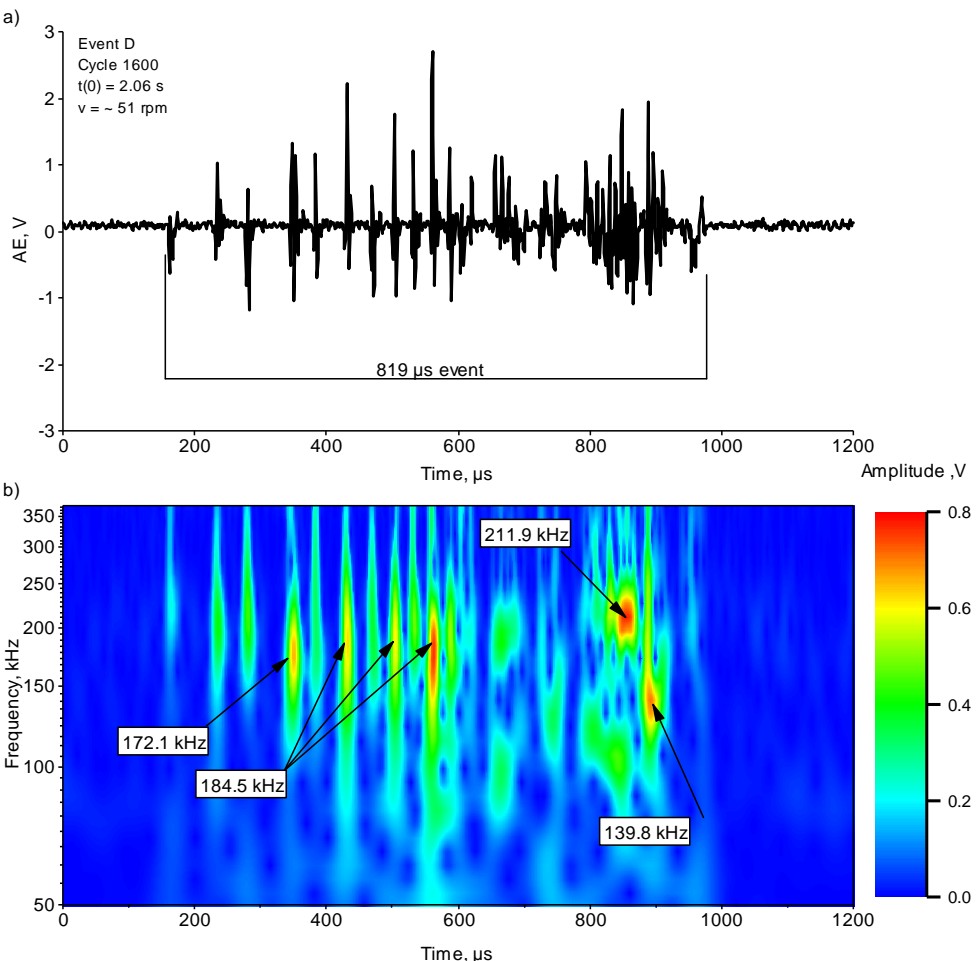

**Figure 8.** Detailed analysis of event **C**: (**a**) unprocessed acoustic emission raw signal of the detected event. (**b**) continuous wavelet transformation (amplitude spectrum) of the event.

The evaluation of this test revealed significant frequency bands of the calculated parameter $P_5$. Peaks of this parameter were found at 86.8 kHz, 137.8 kHz, 206.45 kHz and 231.73 kHz in a 1/12-octave band analysis. In Figure 9, the evolution of the RMS signal in the assigned frequency band is shown over all 9000 cycles and different speeds of the test. In all frequency bands, the first recorded cycles show tendentiously higher values with increasing speed. The RMS values in the 86.8 kHz frequency band seem to tend with increasing number of cycles to lower speeds, as highlighted in Figure 9a. Within the first 1500 to 2000 cycles, this effect is somewhat stronger; presumably this is is due to ongoing adaptation processes. At cycles above 4500, high RMS values are found over the entire speed range, whereby frequencies assigned to adhesive wear mechanism are expected to be found at low speeds. A noticeable finding is the fact that the intensity is not decreasing. On the one hand, the dissipated energy, due to asperity contact, increases naturally with the speed, but on the other hand, some high values are only found at high average speeds. A detailed view on the frequency bands at 137.8 kHz and 231.73 kHz reveals slightly higher values above cycle 1200 and an average speed of 75 rpm. In Figure 9b,d, the similar behavior of the frequency bands can be seen clearly, presumable due to an identical

particle-driven excitation mechanism. The overall values are decreasing up to cycle 5000, presumable some kind of adaptation process is still active. At higher cycles the behavior changes again and the intensity increases with the average speed but decreases frequently above 200 rpm. In Figure 9c, the evolution of the RMS value in the 206.45 kHz band is depicted. At cycles below 1500 the running in behavior is quite similar to the observation in the 86.8 kHz band, but at a slightly higher average speed and over a wider speed range. Starting about at cycle 4000, the behavior changes and high values are found in all speed levels.

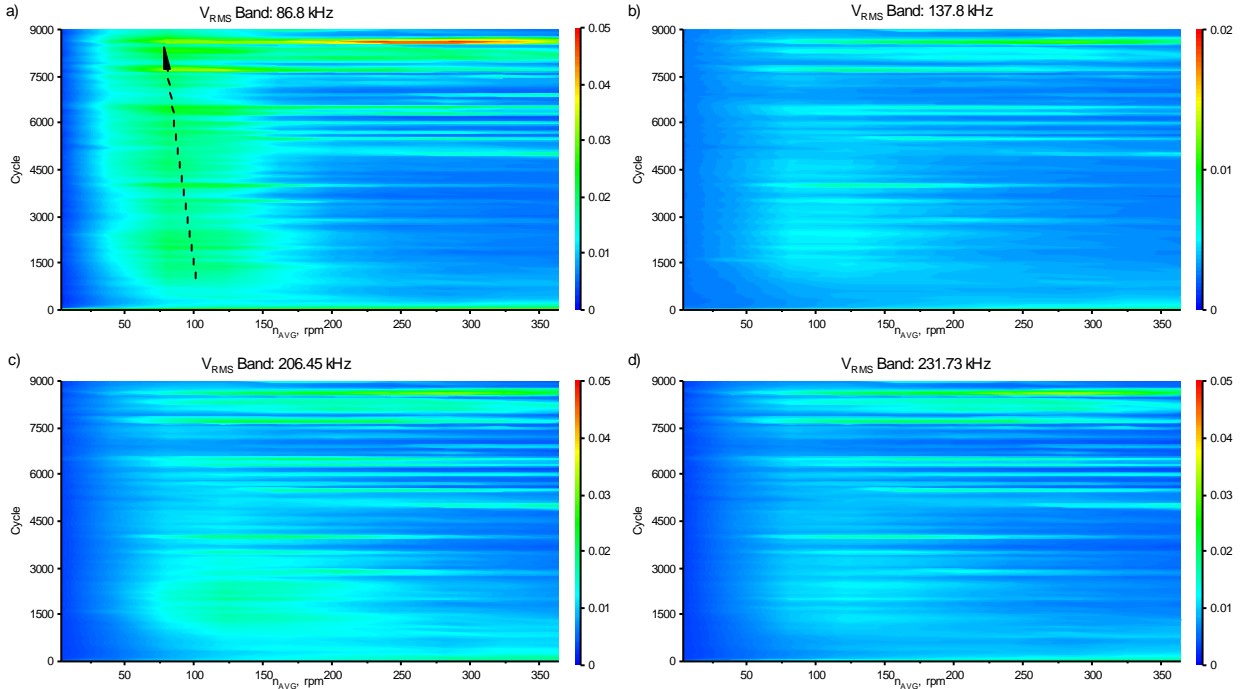

**Figure 9.** Cyclic evolution of the RMS level in certain frequency bands over speed ramps. (**a**) shift of peak values towards lower average speed in the 86.8 kHz Band; (**b**) 137.8 kHz; (**c**) 206.45 kHz Band; (**d**) 231.73 kHz.

The acoustic emission reveals a changing behavior in states of mixed friction, which can be seen in Figure 10. For the illustration four Stribeck curves with previously detected AE events were chosen, unfortunately the observed pulses are to short and weak to be visible in the friction signal. Nevertheless, the evolution of these curves is quite interesting, especially at speeds below 200 rpm in states of mixed friction. In cycle 1600, the coefficient of friction decreases significantly slower than observed in higher cycles. The area of mixed friction is shifted about 50 rpm between early and later cycles. This observation is in coincidence with the 86.8 kHz frequency band, which is attributed to adhesive wear.

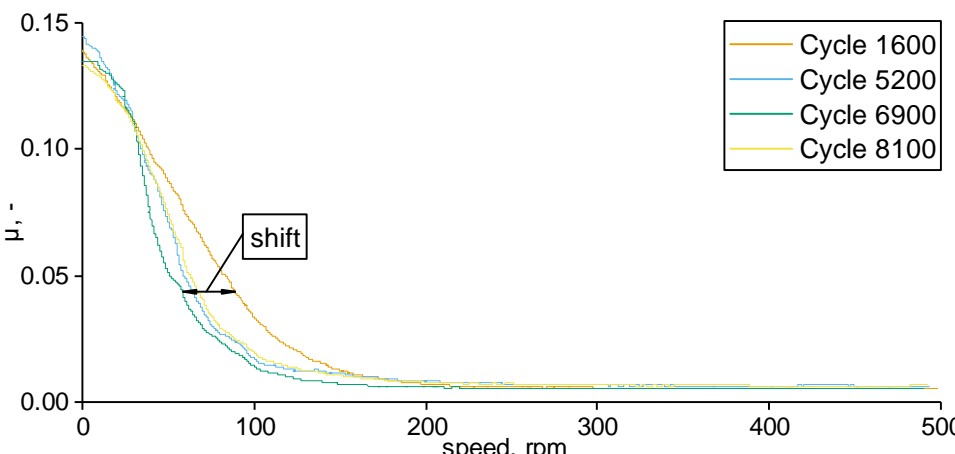

**Figure 10.** Stribeck curves of cycles with detected features. Conspicuous is the varying behavior in mixed friction between cycle 1600 and higher cycles.

### 3.2. Simulative Results

In order to investigate the influence of an artificial groove a simulation was performed. The resulting friction coefficient in dependence of the shaft speed was calculated for different groove depths $w_z$. In Figure 11a, the shapes of different worn Stribeck curves are compared to each other. Generally, increasing static friction coefficients are found at higher values of $w_z$. Since the true contact area rises and the average lubrication decreases, consequently more asperities are in mechanical contact. Apart from the absolute COF value, the boundary friction area changes to a more convex shape, which leads to higher friction values over a broader speed range. The differences between Stribeck curves with a groove compared to virgin bearing segments decreases significantly at higher speed levels. Altered hydrodynamic conditions in the lubrication gap have a significant influence on the lateral displacement. In Figure 11b, the calculated displacement in x-direction of a bearing segment for different values of $w_z$ are compared.

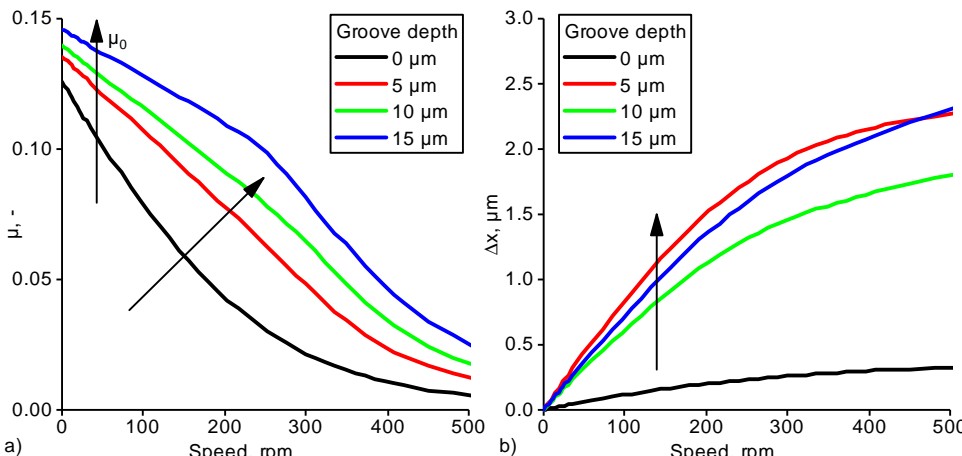

**Figure 11.** Results of the numerical simulation with elastically supported bearing segments. (**a**) Morphology of the Stribeck curve depended on the groove depth. (**b**) Lateral displacement of the bearing segment due to an increasing horizontal force component.

With the aid of the numerical simulation changes on the formation of the hydrodynamic pressure and the contact pressure were found. The results for a bearing segment with an ideal geometry, at different shaft frequencies, are shown in Figures 12a,b. As a consequence of the lateral displacement both pressure distributions are shifted depending on the speed and sense of motion. A deviant behavior was observed for the case of an

artificial groove in Figures 12c,d. Due to a 5 µm groove the formation of the hydrodynamic pressure is disturbed significantly and two peaks are found. A detailed explanation is given in Figure 12c. The geometric edges of the symmetrically positioned groove, are for this case at ±27.9°, implying a fully conformal contact in this area. This fact disrupts the formation of the hydrodynamic fluid film as long as the lateral displacement is low. At higher speeds, according to Figure 11b, the displacement increases and the former conformal gap divides in to a converging and a diverging section. The formation of a load carrying hydrodynamic film is delayed as a direct consequence of the imperfection. These altered conditions have a significant influence to the solid contact, which appears usually at the point of the lowest bearing clearance. This issue is considered by a clearance dependent model for the asperity pressure. The simulation for the worn bearing, see Figure 12d, reveals two areas of high loads just beyond the artificial gap for low speeds and a single peak, in the sense of motion, for higher speeds.

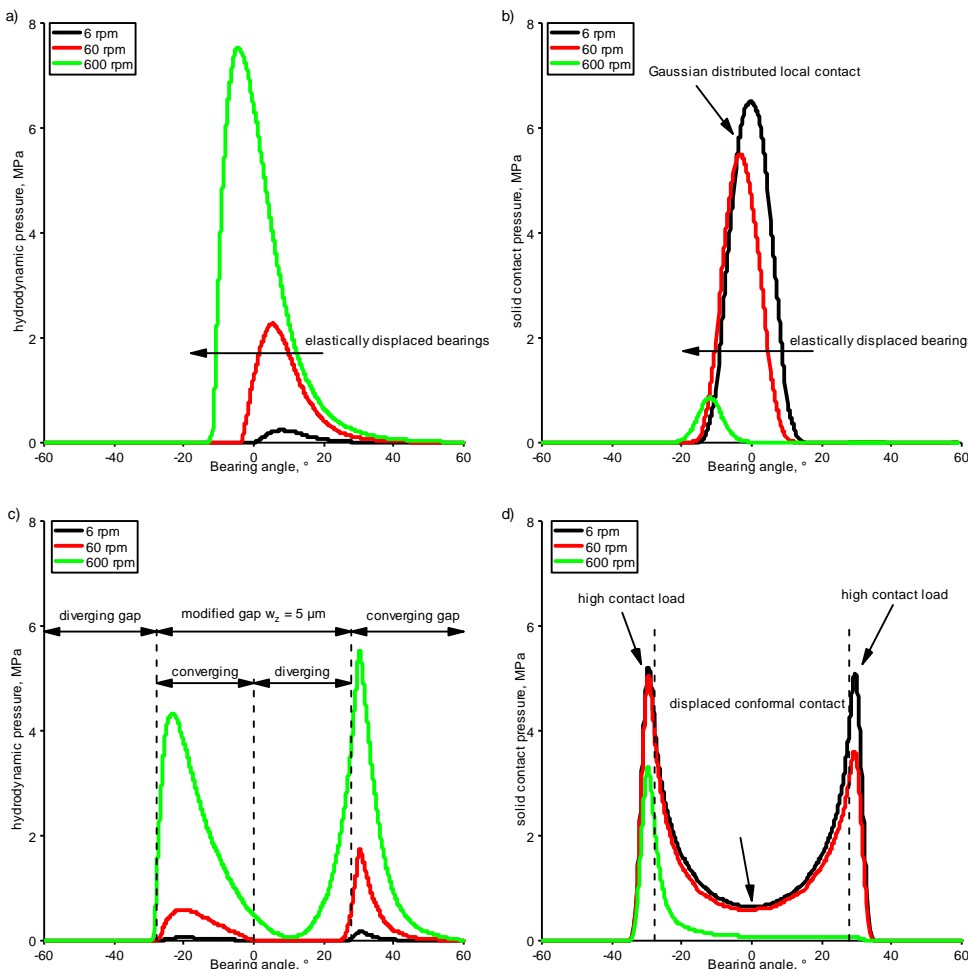

**Figure 12.** Pressure distributions for different shaft frequencies derived from the numerical simulation. Virgin bearing segment: (**a**) hydrodynamic pressure, (**b**) solid contact pressure, bearing segment with an artificial groove: (**c**) hydrodynamic pressure, (**d**) solid contact pressure.

## 4. Discussion

The fact that, in different frequency bands of the transient octave band analysis revealed two wear mechanisms, which could be identified by their characteristic frequency in the acoustic emission spectrum. Hase et al. [3] matched frequencies about 80 kHz with adhesive wear and frequencies about 200 kHz with particles. In the described experiment, high values in the 86.8 kHz and 206.45 kHz frequency bands were found. Considering a journal bearing with an accelerating shaft, it is commonly known that with increasing

speed, the hydrodynamic pressure grows continuously. As a consequence of the changing pressure distribution in the lubrication gap, the resulting force vector changes the direction, which leads to a displacement of the shaft for a fixed journal bearing. Since the reaction force is varying, the motion of the shaft can be described by a curve. However, as soon as the relative speed increases the lubrication gap increases continuously so that the shaft is only at low speeds in a direct solid body contact with the surface of the journal bearing. As a logical consequence of that adhesive wear is expected to happen only at low speed, which was not observed in our experiment. Particles in a tribological system are very common, but are usually hard particles ripped out of the soft matrix of a bearing material. Since silver is very ductile, surprisingly high intensities were found in corresponding frequency ranges. In the following an explanatory approach is stated a visualization of the proposed mechanism is given in Figure 13. The basic idea is the formation of a local groove during the running in procedure as well as at very low speeds in the following speed cycles. As the speed increases cyclically, the groove in the center of the journal bearing is already present. In Figure 13a, the local groove with the radius of the shaft is depicted. On the basis of the ductility of silver it can be assumed that the coating is smeared out of the contact area and re-agglomerates beside. As soon as the shaft accelerates hydrodynamic forces push away the journal bearing in both directions and the agglomerations are overrun and erroded by the shaft, whereby they are released in the shape of small particles. The expected displacement of the bearings in dependence of the sense of rotation is shown in Figure 13b. Some of these soft and ductile particles are retracted by the opposing journal bearing. Both of these connected particle mechanisms happen in a very short time during the acceleration ramp. The observed burst signal in combination with characteristic frequencies in Event **C** can be explained with this theory.

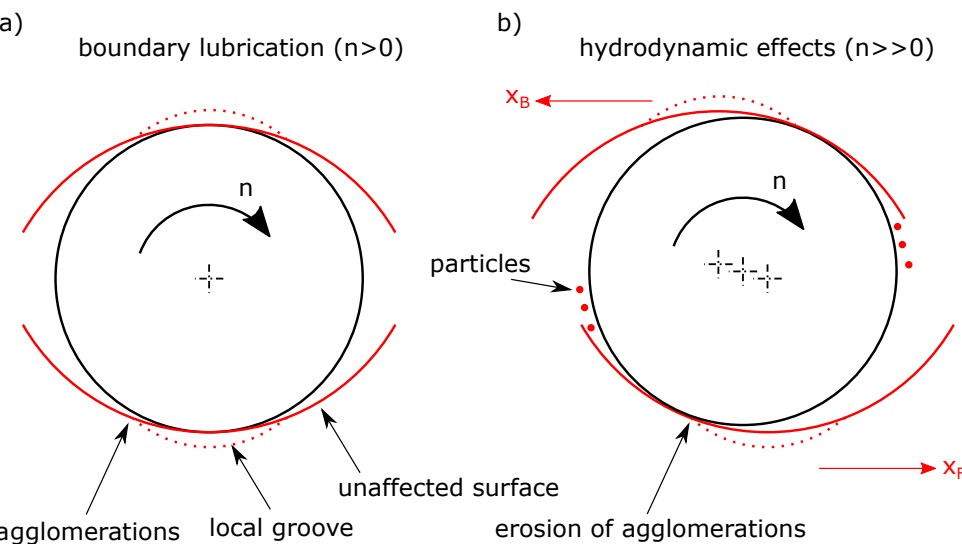

**Figure 13.** Proposed wear mechanism with adhesive wear at low speed and particles at higher speeds. (**a**) agglomeration of deformed silver in high local contact pressure areas at very low speed, (**b**) displacement of the elastically fixed bearing shells due to superimposed hydrodynamic force.

A subsequentially performed numerical simulation revealed a significant change in the contact pressure distribution for different shaft speeds and artificial wear depths $w_z$, which is concurring with the proposed wear mechanism.

## 5. Conclusions

The authors evaluated acoustic emission datasets with derived parameters trimmed for the detection of tribological events. The main findings of our work are:

- The use of the kurtosis, especially in combination with parameters describing the signal intensity is an appropriate tool to detect tribological events in acoustic emission

datasets. An application of the proposed parameter is suitable for an intelligent data acquisition trigger, whereby the trigger can be defined for a special frequency band.

- Short time tribological events like small pulses from particles are emitting a broadband signal, which can be detected easily in most frequencies. Nevertheless, the highest intensities are found to be in certain frequency bands.
- The presence of adhesive wear mechanisms with simultaneously detected particles were successfully detected in a non-stationary operated segment of a journal bearing.
- A theory for a ductile wear mechanism in a silver coating was formulated on the interpretation of a 1/12-octave band analysis on acoustic emission measurements. With a supplementary numerical simulation with appropriate elastic boundary conditions, the behavior could be justified by a speed depended lateral displacement.
- The evolution of the stribeck curve in mixed friction can be explained by the proposed wear mechanism.

**Author Contributions:** Conceptualization, P.R., F.S. and F.G.; data curation, P.R.; investigation, P.R.; methodology, P.R. and F.S.; project administration, F.S. and F.G.; resources, F.G. and A.E.; supervision, F.S.; validation, P.R.; visualization, P.R.; simulation, M.M.; writing—original draft, P.R.; writing—review and editing, F.S., F.G., M.M., C.S. and A.E. All authors have read and agreed to the published version of the manuscript.

**Funding:** This project has received funding from the Clean Sky 2 Joint Undertaking under the European Union's Horizon 2020 research and innovation programme under grant agreement No 785414.

**Conflicts of Interest:** The authors declare no conflict of interest.

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
