# Peer review of "Monitoring Tribological Events by Acoustic Emission Measurements for Bearing Contacts"

_lubricants, doi:10.3390/lubricants9110109_

Round 1

Reviewer 1 Report

The content of this draft is a very interesting result. I think that it is useful in proposing a research method for organizing AE signals.

On the other hand, there are many studies on AE in past papers and literature on the correlation between frequency characteristics and wear and seizure. The correlation between the prediction of wear at the frequency obtained by the AE signal in this study and the form of wear in the previous study is unclear.

The following points are questions and comments.

(1) In this study, the testing machine shown in Fig. 1 is used.

The correlation with Fig. 2 is difficult to understand. It is better to modify the diagram of the experimental equipment to make it easier to understand.

(2) The AE sensor seems to be attached to Bracket, but there are no details on the installation or specifications of the AE sensor.

(3)There is no concrete comparison with this research content compared to the previous research. Previous studies have explained the relationship between wear and seizure phenomena and AE and actual bearing surface observations.

For example, in conventional research

①Plastic deformation: region around 0.1 MHz

②Abrasive wear : around 0.6MHz

③Adhesive wear : around 1.4MHz

This draft mainly ranges from 0.05MHz (50kHz) to 0.2MHz (200kHz), and describes the correlation between the calculated lubrication region and the AE signal. For this reason, please add a comparison with this high frequency region and wear phenomenon.

(Especially, Abrasive wear range around 0.6MHz)

(4) In this draft, the surface is simulated by calculation, but how did you confirm the correlation with the surface of the test bearing?(For example, secondary electron image, etc.)

Author Response

Dear Reviewer,

thank you for your constructive review and annotations. Please see the attachment.

Regards

Reviewer 2 Report

The authors have done great work.  The article is written very well, concisely, and can be accepted in the present form.

Author Response

Dear Reviewer,

thank you for your motivating response. Please see the attachment.

Regards

Reviewer 3 Report

Dear Authors,

You wrote an interesting paper in an scientifically relevant field. 

However, some data and reflections are missing in your paper:

  1. What was the initial surface roughness of your components? And after the tests?
  2. The wear mechanism you proposed is interesting, but the particle-wear apparition at higher speeds might just be due to mixing (perturbing) of the oil bath at higher speeds.
  3. Please try to provide an explanation for the 86.8 - 137.8 - 201 kHz? Why these frequencies? Are you sure this is not related to a test rig component noise? Was this (or something similar) seen in the literature?
  4. Is the shape of the bearing & journal (after the tests) correlated to your wear simulations? Could you really see the proposed abrasive and adhesive wear regions, as well as the wear rates that you simulated? This would be very important for the scientific value of your paper.

Author Response

(The authors gave the same response as above.)

Round 2

Reviewer 1 Report

It is very useful to grasp the lubrication state by the AE signal. This research result is disappointing because there are frequency bands where signal processing cannot be performed. It will be very useful if sampling and signal processing up to about 1.4MHz can be performed in the future. I believe that the content of this draft will be useful to many readers as a method of processing AE signals.